# OpenReview forum: "Disentangling Intent from Role: Adversarial Self-Play for Persona-Invariant Safety Alignment"
_ICML.cc/2026/Conference — ICML 2026 regular_

### Official Review · Reviewer_PuM9 · 2026-02-23

**Soundness:** 2
**Presentation:** 4
**Significance:** 3
**Originality:** 3
**Overall Recommendation:** 5
**Confidence:** 4

**Summary:**

This paper focuses on persona-based jailbreak attacks, a specific type of adversarial attack. The main research question is whether an LLM’s safety decision should consider persona context, and they proposed the structural separation hypothesis to answer this question. The authors also proposed PLE as a systematic approach to selecting the strongest persona to maximise ASR, and PICL as a form of defense that can decouple safety decisions from persona contexts.

**Compliance With Llm Reviewing Policy:**

Affirmed.

**Final Justification:**

The rebuttal addressed my main concerns, so i have increased the score to 5.

**Key Questions For Authors:**

1. Can you provide more experiment results on different LLMs? I am interested in the transferability and the robustness of PLE and PICL as shown in Tables 1 and 2, as the current results are very strong.
2. Can you discuss the underlying premises and provide justification for the methodological choices? For example, it will be great if you can briefly discuss what assumptions you made in order to use the variational properties of mutual information to obtain a variational upper bound in lines 124-141.
3. Is there any other sensitivity analysis you can provide to prove the implications of hypothesis 1 (lines 144-146)? DPO and SFT are known to be very strong LLM defense methods, and Table 5 shows that they do significantly drop the ASR of attacks. Is it possible to solely assess the significance of PIC?

**Limitations:**

yes

**Strengths And Weaknesses:**

Strengths:

1. The best part of this paper is the structural separation hypothesis. I think this is a very interesting property to look into, as it explores the relationship between safety behaviour, harmful intention, and persona context. I think there is potential for research in these relationships.

2. The methodologies proposed are also innovative. I really like the idea of PICL that tries to decouple safety decisions from persona context. I also think that it's novel for PLE to treat persona attack as a graph-based optimization problem. I am not entirely familiar with the lineage-amnesia problem in the adversarial attack setting, but it seems important, and the authors' proposed approach seems clever.

3. Both PLE and PICL are simple enough to obtain significant improvement from the existing methods. PLE is basically an optimised search algorithm based on “Enhancing jailbreak attacks on llms via persona prompts”, while PICL is largely based on DPO and SFT. However, both methods achieved performance improvement on their respective aspects.

Weaknesses:

1. Justification of methodological choices and underlying premises is required. One thing that is consistently lacking throughout the paper is the premises made to select certain methods. For example, in lines 125-140, it is understandable that we cannot do the direction optimization. However, I am confused about why we use the variational properties of mutual information to obtain a variational upper bound. What premises or assumptions did we make so that we can use the inequality shown in line 136? The same problem occurred when the authors formulated persona search as a graph optimization problem and proposed PLE.

2. The core research problem, the structural separation hypothesis (especially the implications in lines 144-146), and the experiments shown, appear loosely connected. In the experiment section, I expected a sensitivity analysis of the adversarial attack's dependency on persona context. Moreover, the experiments showed that the ASR does drop after applying PICL, but then PICL is largely based on DFO and SFT, which are trivial methods known to be highly effective in LLM defence. Even as shown in Table 5, the ASR dropped significantly with the DPO and SFT only. I am not convinced that the structural separation hypothesis and its implications are actually proved in this paper.

3. More experiments across different LLMs are required to further consolidate the methodology's significance. There are only two models (Qwen 2.5 and Llama 3.1) evaluated in section 5. It would be great if the authors could include more LLMs for evaluation.

---

> ### Author Rebuttal · Authors · 2026-03-31
>
> Thank you for your insightful comments. We have carefully addressed each of your concerns below and sincerely hope that our responses are helpful.
>
> ### **Weakness 1 & Question 2**
>
> **Mutual information formulation.**
>
> We require only that the auxiliary distribution $\rho(\cdot|q)$ be independent of the persona variable $p$.
>
> Since the marginal distribution $\pi\_\theta(y|q)\_{\mathrm{marg}}$ in $I(Y; P|q)$ is intractable to compute, we leverage a classic variational formulation to derive a tractable surrogate objective. Let $n\_p(y):=\pi\_\theta(y|p,q)$ and $m(y):=\pi\_\theta(y|q)\_{\mathrm{marg}} =\mathbb{E}\_{p\sim\mathcal{P}}[n\_p(y)]$, we have
> $$
> \begin{equation}
> D_{\mathrm{KL}}(n_p\|m)=\sum_y n_p(y)\log\frac{n_p(y)}{m(y)}=\sum_y n_p(y)\log\frac{n_p(y)}{\rho(y)}-\sum_y n_p(y)\log\frac{m(y)}{\rho(y)}=D_{\mathrm{KL}}(n_p\|\rho)-D_{\mathrm{KL}}(m\|\rho)-\sum_y\bigl(n_p(y)-m(y)\bigr)\log\frac{m(y)}{\rho(y)}.
> \end{equation}
> $$
> When $\rho(\cdot|q)$ is independent of $p$, taking expectation over $p\sim\mathcal{P}$, since the last term $\mathbb{E}\_{p\sim\mathcal{P}}[\sum\_{y}(n(y)-m(y))\log\frac{m(y)}{\rho(y)}]=\sum\_{y}(\mathbb{E}\_{p}[n(y)]-m(y))\log\frac{m(y)}{\rho(y)}=0$ and KL divergence is non-negative, we obtain
> $$
> I(Y;P|q)\le\mathbb{E}\_{p\sim\mathcal{P}}[D\_{\mathrm{KL}}(\pi_\theta(\cdot|p,q)\,\|\rho(\cdot|q))].
> $$
> When the output distributions $\pi_\theta(\cdot|p,q)$ under all persona contexts are aligned toward a common persona-free distribution $\rho(\cdot|q)$, the dependence of $Y$ on $P$ is correspondingly reduced.
>
> **PLE as graph optimization.**
>
> The design of PLE does not rely on additional assumptions. It follows from the need to improve genetic algorithms in addressing the well-known issue of evolutionary amnesia. This formulation enables efficient discovery of high-risk personas while preserving long-term evolutionary signal.
>
> ### **Weakness 2 & Question 3**
> Our work establishes a coherent hypothesis–mechanism–validation pipeline. The structural separation hypothesis, motivated by observations of persona-based jailbreak attacks, formalizes that safety-critical decisions should remain invariant under persona perturbations. We quantify persona-based distribution shift via conditional mutual information, and use a variational upper bound to obtain a tractable objective, yielding the PICL objective. PIC cannot be evaluated in isolation, as it is insufficient to learn explicit safety preference structures.
>
> We further validate the hypothesis by analyzing the standard deviation of ASR across 35 OOD elite personas in **Table 1**. The results show that PICL enables persona-invariant safety behavior: under the w/o-PIC setting, ASR varies substantially across personas for the same harmful instruction, whereas with full PICL, ASR remains consistently stable across personas.
>
> **Table 1. The standard deviation of ASR across 35 OOD elite personas.**
> |Model|Method|StrongREJECT|XSTest-contrast|Malicious.|
> |-|-|-|-|-|
> |Qwen2.5-7B-Instruct|+w/o PIC|0.0260|0.0952|0.0534|
> ||**+ PICL (Ours)**|0.0067|0.0083|0.0041|
> |Llama-3.1-8B-Instruct|+w/o PIC|0.0834|0.1084|0.0752|
> ||**+ PICL (Ours)**|0.0057|0.0064|0.0022|
>
> The ablation results also validate the hypothesis. By aggregating results in Table 2 (see supplementary link: https://anonymous.4open.science/r/anonymous-result), we observe that PICL consistently reduces persona-based failures compared to DPO+SFT, while preserving stronger utility. These findings demonstrate that PICL contributes additional robustness specifically against persona-based distributional shifts.
>
> ### **Weakness 3 & Question 1**
> We further strengthen the empirical validity of our method by extending the evaluation to a third backbone, Mistral-7B-Instruct-v0.3 [1], with conclusions that are fully consistent with our main claims.
>
> As shown in **Table 3**, PLE demonstrates strong cross-benchmark transferability, driving ASR to near saturation across benchmarks. This confirms that PLE discovers highly transferable adversarial personas rather than model-specific artifacts. As shown in **Table 4**, PICL remains highly effective on Mistral, achieving large and consistent ASR reductions. This indicates that the robustness induced by persona-invariant consistency generalizes beyond Qwen and Llama to an additional, architecturally distinct LLM family.
>
> **Table 3. Attack transferability of elite personas on OOD harmful benchmarks.**
> |Method|SafeRLHF-unsafe|StrongREJECT|DAN|Harmbench|OR-toxic|
> |-|-|-|-|-|-|
> |Mistral-7B-Instruct-v0.3|0.352|0.390|0.401|0.523|0.315|
> |**+ PLE (ours)**|0.954|0.968|0.961|0.983|0.985|
>
> **Table 4. Robustness against OOD persona-based jailbreak attacks.**
> |Method|StrongREJECT|WildGuardTest|XSTest-contrast|Harmbench|
> |-|-|-|-|-|
> |Mistral-7B-Instruct-v0.3|0.786|0.390|0.445|0.685|
> |**+ PICL (ours)**|0.019|0.029|0.016|0.025|
>
> **Reference**
>
> [1] URL https://huggingface.co/rubra-ai/Mistral-7B-Instruct-v0.3.

---

> > ### Author Rebuttal · Reviewer_PuM9 · 2026-04-03
> >
> > Thank you for the clarifications and for providing additional experiments. I appreciate the effort and I will consider increasing the score to 5!
> >
> > P.S. Please consider evaluating on newer models next time. Mistral 7B is like 3 years ago.

---

> > > ### Author Response · Authors · 2026-04-04
> > >
> > > Thank you for your positive feedback and for acknowledging our clarifications. We are encouraged by your willingness to raise the evaluation to 5!
> > >
> > > In response to your suggestion to evaluate newer models, we have extended our evaluation to gemma-3-4b-it [1], an advanced lightweight instruction-tuned model released in March 2025. These additional experiments reinforce our main claims and provide consistent support for our conclusions.
> > >
> > > As shown in **Table 1**, the target model (gemma-3-4b-it) demonstrates safety under direct queries, with base ASRs remaining below 10% across most datasets. In contrast, elite personas evolved by PLE significantly degrade safety, exhibiting strong cross-benchmark transferability and driving ASR to near saturation across multiple datasets. As shown in **Table 2**, PICL remains highly effective on Gemma, significantly mitigating persona-based jailbreak attacks and achieving consistent and substantial reductions in ASR. This demonstrates that the robustness provided by persona-invariant consistency generalizes beyond Qwen and Llama, extending to another LLM family.
> > >
> > > **Table 1. Attack transferability of *elite personas* on OOD harmful benchmarks.**
> > > |Method|SafeRLHF-unsafe|StrongREJECT|WildGuardTest|XSTest-contrast|Advbench|DAN|Harmbench|Malicious.|OR-toxic|
> > > |-|-|-|-|-|-|-|-|-|-|
> > > |gemma-3-4b-it|0.011|0.026|0.082|0.000|0.010|0.312|0.163|0.000|0.036|
> > > |**+ PLE (ours)**|0.957|0.965|0.355|0.895|0.964|0.503|0.778|0.990|0.940|
> > >
> > > **Table 2. Robustness against OOD persona-based jailbreak attacks.**
> > > |Method|StrongREJECT|WildGuardTest|XSTest-contrast|DAN|Harmbench|MaliciousInstruct|OR-Bench-toxic|
> > > |-|-|-|-|-|-|-|-|
> > > |gemma-3-4b-it|0.859|0.334|0.480|0.432|0.655|0.940|0.779|
> > > |**+ PICL (ours)**|0.032|0.027|0.019|0.155|0.042|0.009|0.063|
> > >
> > > **Reference**
> > >
> > > [1] URL https://huggingface.co/google/gemma-3-4b-it.

---

### Official Review · Reviewer_XwQq · 2026-03-10

**Soundness:** 4
**Presentation:** 4
**Significance:** 3
**Originality:** 3
**Overall Recommendation:** 5
**Confidence:** 3

**Summary:**

This paper studies the persona-based jailbreaking problem of LLM alignment.
The writers introduce a theoretical framework that captures how modern LLMs fall for the persona based jailbreaking method and how an ideal LLM should behave as persona-invariant under such attacks. The analysis views the vulnerability to persona based attacks as the mutual information between the generated response and the invoked persona, and bounds it in terms of the the KL divergence between the model distribution and a persona invariant distribution.

This bound inspires the "persona-invariant alignment", an experimental framework based on an adversarial self-play, in which an "attacker" learns to generate personas that invoke misalignment in a "defender" (the LLM that is being aligned), while the defender learns to answer queries in a persona invariant manner.
The writers show the personas generated by the attacker of this framework are a powerful tool for jailbreaking, and that the resultant defender exhibits robust alignment on a variety of jailbreaking benchmarks relative to the base model and a DPO and SFT trained variants of the base model. All while keeping the utility of the model intact (namely - lack of over refusal, and capabilities of reasoning and knowledge QA benchmarks).

**Compliance With Llm Reviewing Policy:**

Affirmed.

**Final Justification:**

The writers have addressed my concerns.

**Key Questions For Authors:**

Questions:
- Is persona invariance on benign instructions countered in any way by the training process? Is this behavior exhibited on the resultant model? Is there a way to handle this?

**Limitations:**

yes

**Strengths And Weaknesses:**

Strengths:
- Significance of the problem - persona-based jailbreaking is an important aspect of LLM safety.
- Presentation is clear.
- The notion of persona-invariant alignment is compelling and seems like an interesting approach to the persona-based jailbreaks. The theoretical analysis provides a principled motivation.
- The adversarial self-play framework for alignment is original.
- The personas learned through these frameworks are by themselves an interesting jailbreaking method, as shown on the base models.
- The defender LLM learned through this framework outperforms DPO trained and SFT trained variants of the base model on a large number of jailbreaking datasets, showing the alignment extends beyond the scope of the personas of the framework.
- Utility remains intact - over refusal and performance on reasoning benchmarks remains comparable to the DPO and SFT baselines.


Weaknesses:
- While persona invariance is a good way to overcome persona-based jailbreaks, it harms a useful capability of LLMs - their ability to provide answers based on impersonation of prompted roles. I may have missed how this is handled in the paper, but I imagine persona invariance only needs to be invoked on harmful instructions and not on benign instructions. The empirical results show over refusal and reduced useful capabilities do not appear in the trained model due to the L_SFT and L_DPO terms of the loss. But I could not understand if these terms counter persona invariance on benign instructions.

---

> ### Author Rebuttal · Authors · 2026-03-30
>
> Thank you for your insightful comments. We have carefully addressed each of your concerns below and sincerely hope that our responses are helpful.
>
> ### **Weakness & Question**
>
> We clarify that the persona-invariant consistency objective in PICL is designed to stabilize safety decisions under harmful intent, rather than suppress stylistic or lexical variation in benign role-playing.
>
> We took explicit measures during training to prevent the model from learning the spurious correlation that persona implies harmfulness. Specifically, one-third of SFT samples are regenerated under random personas using the same prompt format as in the attack setting. Detailed dataset construction is provided in Appendix C.2. This ensures that persona signals remain decoupled from safety labels.
>
> We further evaluate PICL on two role-playing benchmarks, CharacterEval [1] and RoleBench [2]. The results in **Table 1** and **Table 2** show that models aligned with PICL maintain comparable benign role-playing utility to the base model. On CharacterEval, performance remains nearly unchanged: for Qwen2.5-7B-Instruct, Character Consistency shifts from 2.833 to 2.823, Conversational Ability from 3.645 to 3.631, and Role-playing Attractiveness remains stable; similar marginal changes are observed for Llama-3.1-8B-Instruct. On RoleBench, PICL largely preserves performance, with Rouge-L F1 slightly improving from 0.172 to 0.174 on Qwen2.5-7B-Instruct and changing marginally from 0.164 to 0.161 on Llama-3.1-8B-Instruct.
>
> **Table 1. Performance on the CharacterEval benchmark (higher is better).** *Notes*: CC = Character Consistency; CA = Conversational Ability; RA = Role-playing Attractiveness.
> |Method|KE|KA|KH|PB|PU|CC avg.|Flu.|Coh.|Cons.|CA avg.|HL|CS|ED|Emp.|RA avg.|
> |-|-|-|-|-|-|-|-|-|-|-|-|-|-|-|-|
> |Qwen2.5-7B-Instruct|2.312|2.915|2.933|3.008|2.998|2.833|3.535|3.879|3.520|3.645|3.268|3.346|2.392|3.202|3.054|
> | **+ PICL (Ours)**|2.317|2.880|2.944|2.984|2.987|2.823|3.468|3.892|3.534|3.631|3.245|3.376|2.392|3.202|3.054|
> |Llama-3.1-8B-Instruct|2.036|2.756|2.709|3.104|2.855|2.692|3.259|3.722|3.409|3.463|3.194|2.848|2.552|2.844|2.860|
> | **+ PICL (Ours)**|1.945|2.731|2.631|3.068|2.785|2.632|3.278|3.627|3.396|3.434|3.190|2.732|2.534|2.803|2.815|
>
> **Table 2. Performance on the RoleBench benchmark. We report RougeL scores.**
> |Method|Recall|Precision|F1|
> |-|-|-|-|
> |Qwen2.5-7B-Instruct|0.125|0.314|0.172|
> | **+ PICL (Ours)**|0.125|0.322|0.174|
> |Llama-3.1-8B-Instruct|0.120|0.291|0.164|
> | **+ PICL (Ours)**|0.119|0.291|0.161|
>
> **Reference**
>
> [1] Charactereval: A chinese benchmark for role-playing conversational agent evaluation. ACL 2024.
>
> [2] Rolellm: Bench-marking, eliciting, and enhancing role-playing abilities of large language models. ACL 2024.

---

> > ### Author Rebuttal · Reviewer_XwQq · 2026-04-03
> >
> > I thank the writers for their response to my concerns. Following this I have raised my score.

---

> > > ### Author Response · Authors · 2026-04-04
> > >
> > > We sincerely appreciate your encouraging follow-up and are delighted to hear that our rebuttal has helped clarify the paper. We are also deeply grateful for your positive assessment and for recognizing the value of our clarifications.

---

### Official Review · Reviewer_DDWm · 2026-03-13

**Soundness:** 2
**Presentation:** 3
**Significance:** 2
**Originality:** 3
**Overall Recommendation:** 3
**Confidence:** 4

**Summary:**

The paper proposes an adversarial self-play framework named Persona-Invariant Alignment (PIA) to defend against persona-based jailbreak attacks. On the attack side, the authors propose Persona Lineage Evolution (PLE), which models persona search as a directed acyclic graph  and integrates lineage-based credit propagation and an Upper Confidence Bound (UCB) mechanism  into a genetic algorithm to enhance exploration efficiency. On the defense side, the authors propose Persona-Invariant Consistency Learning (PICL). This method utilizes a forward KL divergence constraint between the persona-free distribution $\pi_\theta(\cdot|q)$ (acting as a stop-gradient teacher) and the persona-conditioned distribution $\pi_\theta(\cdot|p,q)$ to enforce safety consistency under contextual perturbations, where $p$ and $q$ denote the given prompt and persona, respectively.

**Compliance With Llm Reviewing Policy:**

Affirmed.

**Key Questions For Authors:**

- Can the authors provide evaluation results on standard role-playing benchmarks (e.g., RoleBench, CharacterEval) for the PICL-aligned models?

**Limitations:**

The authors adequately discuss their empirical results but fail to acknowledge the broader limitations of output-level alignment paradigms. A critical limitation is the absence of exploration into deep alignment methods. Rather than relying on superficial probability penalties, future work should discuss or incorporate parameter-level concept erasure, representation engineering, or utilizing "System 2" introspective reasoning capabilities (such as the STAIR framework mentioned in the related work ) to construct more generalized and resilient safety mechanisms.

**Strengths And Weaknesses:**

Strengths:
- Significance & Relevance: The submission addresses a highly critical and timely vulnerability in aligned Large Language Models (LLMs)—persona-based jailbreak attacks. As LLMs are increasingly deployed in interactive and role-playing applications, systematically defending against structurally disguised intents is of great importance to the community.
- Presentation: The paper is well-structured and clearly written. The authors provide a coherent information-theoretic perspective (conditional mutual information) to formalize the mechanism of persona-based attacks and successfully map this theory to their proposed variational upper bound in the defense strategy.

Weaknesses:
- Marginal Methodological Improvement: While the PICL objective is theoretically framed around a "structural separation hypothesis", its implementation is functionally equivalent to automated self-distillation or consistency regularization . By minimizing the forward KL divergence $D_{KL}(sg[\pi_{\theta}(\cdot|q)] || \pi_{\theta}(\cdot|p,q))$, the gradient updates essentially act as a cross-entropy penalty that forces the model to use its own persona-free safe responses as pseudo-labels. While empirically effective at reducing the Attack Success Rate (ASR), this acts more as an engineering heuristic rather than a significant improvement in alignment.
- Susceptibility to "Shallow Alignment" : Literature highlights that target-distribution matching techniques (such as SFT and DPO ) often result in shallow alignment, where models merely learn to output conditional refusal tokens at the surface level without altering the deep internal representations of harmful concepts. Because PICL only operates on the final output logits, it is highly likely that the model suffers from the same superficial alignment.
- Compromised Role-Playing Utility: By definition, the PICL objective forces the persona-conditioned output to strictly align with the persona-free baseline. While this patches jailbreak vulnerabilities, it theoretically destroys the model's ability to maintain high fidelity in legitimate, benign role-playing scenarios (e.g., adopting a specific tone, style, or vocabulary). The authors claim that "general capability" is preserved based on standard QA benchmarks like MMLU and GPQA. However, evaluating mathematical and reasoning abilities is entirely orthogonal to evaluating persona fidelity.

---

> ### Author Rebuttal · Authors · 2026-03-30
>
> Thank you for your insightful comments. We have carefully addressed your concerns below and would appreciate your reconsideration if you find them adequately addressed.
>
> ### **Weakness 3 & Question**
> We further evaluate PICL on two role-playing benchmarks, CharacterEval [1] and RoleBench [2]. The results in **Table 1** and **Table 2** show that models aligned with PICL maintain comparable benign role-playing utility to the base model.
>
> On CharacterEval, performance remains nearly unchanged: for Qwen2.5-7B-Instruct, Character Consistency shifts from 2.833 to 2.823, Conversational Ability from 3.645 to 3.631, and Role-playing Attractiveness remains stable; similar marginal changes are observed for Llama-3.1-8B-Instruct. On RoleBench, PICL largely preserves performance, with Rouge-L F1 slightly improving from 0.172 to 0.174 on Qwen2.5-7B-Instruct and changing marginally from 0.164 to 0.161 on Llama-3.1-8B-Instruct.
>
> This behavior is consistent with our training design: we explicitly prevent the model from learning the spurious correlation that persona implies harmfulness. Specifically, one-third of SFT samples are regenerated under random personas using the same prompt format as in the attack setting, ensuring that persona signals remain decoupled from safety labels.
>
> **Table 1. Performance on the CharacterEval benchmark (higher is better).** *Notes*: CC = Character Consistency; CA = Conversational Ability; RA = Role-playing Attractiveness.
> |Method|KE|KA|KH|PB|PU|CC avg.|Flu.|Coh.|Cons.|CA avg.|HL|CS|ED|Emp.|RA avg.|
> |-|-|-|-|-|-|-|-|-|-|-|-|-|-|-|-|
> |Qwen2.5-7B-Instruct|2.312|2.915|2.933|3.008|2.998|2.833|3.535|3.879|3.520|3.645|3.268|3.346|2.392|3.202|3.054|
> | **+ PICL (Ours)**|2.317|2.880|2.944|2.984|2.987|2.823|3.468|3.892|3.534|3.631|3.245|3.376|2.392|3.202|3.054|
> |Llama-3.1-8B-Instruct|2.036|2.756|2.709|3.104|2.855|2.692|3.259|3.722|3.409|3.463|3.194|2.848|2.552|2.844|2.860|
> | **+ PICL (Ours)**|1.945|2.731|2.631|3.068|2.785|2.632|3.278|3.627|3.396|3.434|3.190|2.732|2.534|2.803|2.815|
>
> **Table 2. Performance on the RoleBench benchmark. We report RougeL scores.**
> |Method|Recall|Precision|F1|
> |-|-|-|-|
> |Qwen2.5-7B-Instruct|0.125|0.314|0.172|
> | **+ PICL (Ours)**|0.125|0.322|0.174|
> |Llama-3.1-8B-Instruct|0.120|0.291|0.164|
> | **+ PICL (Ours)**|0.119|0.291|0.161|
>
> **Reference**
>
> [1] Charactereval: A chinese benchmark for role-playing conversational agent evaluation. ACL 2024.
>
> [2] Rolellm: Bench-marking, eliciting, and enhancing role-playing abilities of large language models. ACL 2024.
>
> ### **Weakness 1**
> PICL is not a heuristic, but a principled objective derived from an information-theoretic analysis of persona-based safety failures. Our work establishes a coherent hypothesis–mechanism–validation pipeline.
>
> We begin with the structural separation hypothesis, which formalizes that safety-critical decisions should remain invariant under persona perturbations. We then quantify persona-based distribution shift via the conditional mutual information $I(Y; P \mid q)$, capturing the dependence of model outputs on persona context. Based on this, we derive a variational upper bound and translate it into a tractable training objective by aligning persona-based outputs with a persona-free reference distribution. In practice, we instantiate $\rho$ as $\pi\_\theta(\cdot \mid q)$, a natural approximation of persona-invariant safe behavior, while remaining one valid choice within a broader framework.
>
> Therefore, our method is fundamentally principled. More importantly, this framework not only offers a principled interpretation of self-distillation-style methods but also generalizes beyond them through alternative choices of persona-free reference distributions.
>
> ### **Weakness 2**
> We acknowledge that "shallow alignment" is a concern in LLM safety research. However, our primary focus is on the underexplored and critical problem of persona-based jailbreak attacks, rather than exploring shallow alignment and deep alignment mechanisms.
>
> On the attack side, we propose PLE, which novelly formulates persona attacks as a graph-based optimization problem, addressing the lineage amnesia issue in genetic algorithms and enabling efficient exploration of high-risk persona spaces. On the defense side, we systematically explore the relationship among safety behavior, harmful intent, and persona context, and derive a defense objective.
>
> Empirically, our method shows promising signs of going beyond shallow alignment, as reflected in its strong generalization to unseen persona-based attacks, a clear advantage over existing methods. Moreover, it achieves a better safety–utility trade-off, maintaining benign compliance while avoiding excessive refusal.
>
> In future work, we will integrate deep alignment methods with our framework to construct more generalized safety mechanisms.

---

> > ### Author Rebuttal · Reviewer_DDWm · 2026-04-04
> >
> > Thank the authors for the detailed rebuttal and the additional experiments.
> >
> > 1. Preservation of Role-Playing Utility
> >
> > I sincerely appreciate the effort taken to evaluate the PICL-aligned models on the CharacterEval and RoleBench benchmarks. The supplementary results are convincing and effectively alleviate my initial concerns regarding the potential degradation of benign role-playing capabilities.
> >
> > 2. Concerns on Methodological Novelty
> >
> > However, despite the thorough response, my reservations regarding the methodological novelty remain. While I acknowledge the elegant information-theoretic formulation presented by the authors, the core mechanism fundamentally reduces to output distribution alignment—specifically, regularizing the harmful persona-conditioned distribution to match the safe persona-free baseline. The theoretical framing provides a neat perspective, but the practical implementation operates largely as a standard consistency constraint. In the current alignment landscape, there are already numerous well-established tools and regularization techniques capable of achieving this form of distribution matching. Consequently, the approach feels more like an application of existing distillation/regularization heuristics to a new scenario, rather than a fundamental algorithmic breakthrough.
> >
> > 3. The Issue of "Shallow Alignment"
> >
> > Furthermore, regarding the discussion on "shallow alignment," I understand the authors' boundary that their primary focus is on mitigating persona-based jailbreaks rather than representation engineering. Nevertheless, relying entirely on logits-level correction inherently limits the depth of the safety intervention. As observed in broader safety literature, purely output-centric penalties often teach the model to merely suppress refusal tokens at the surface level without altering the underlying representations of harmful concepts. I remain skeptical about the long-term robustness of such probability-based regularization, as models constrained only at the output layer often remain vulnerable to more sophisticated or deeper representation-level attacks.
> >
> > Overall, I acknowledge that the paper addresses a timely and critical vulnerability, and the empirical defense success rate is commendable. However, due to the foundational reliance on surface-level distribution alignment and limited methodological novelty, I will maintain my current score.

---

> > > ### Author Response · Authors · 2026-04-08
> > >
> > > Thank you for your thoughtful follow-up and additional comments. We appreciate your recognition that our results address concerns about preserving benign role-playing abilities, as well as your acknowledgment of the importance of the problem and the strength of our empirical defense.
> > >
> > > Our primary goal is to address the underexplored problem of defending against persona-based jailbreaks. We contribute a novel persona search framework on the attack side and a principled information-theoretic formulation on the defense side, together advancing this line of research.
> > >
> > > We also thank you for your thoughtful perspective on shallow alignment. Your comments provide valuable guidance for our future investigation of deeper alignment methods.

---

### Official Review · Reviewer_6JXK · 2026-03-14

**Soundness:** 4
**Presentation:** 4
**Significance:** 3
**Originality:** 2
**Overall Recommendation:** 4
**Confidence:** 4

**Summary:**

Overall, this work explores the safety alignment issues of large language models when faced with role-based jailbreak attacks. It proposes a framework called Persona-Invariant Alignment (PIA) as an adversarial self-play mechanism to enhance the model's safety in high-risk scenarios.

**Compliance With Llm Reviewing Policy:**

Affirmed.

**Ethical Review Flag:**

Flag this paper for an ethics review.

**Ethics Expertise Needed:**

["Discrimination / Bias / Fairness Concerns"]

**Final Justification:**

Weak accept.

The authors' rebuttal resolve my concern.

**Key Questions For Authors:**

See weakness.

If the authors can address the concerns regarding comparative methods, I would be willing to raise my score.

**Limitations:**

See weakness.

**Strengths And Weaknesses:**

### Strengths:
1. The work addresses an important issue: how large models can maintain safety stability under different role settings. As AI increasingly trends toward personalized assistants, this question becomes even more critical. Therefore, this work represents a valuable attempt to tackle a significant future problem.

2. The workflow design of this problem is complete and clear. From designing a well-thought-out attack method to developing defenses against that attack, the work is supported by robust theoretical foundations and detailed formulas, along with corresponding experimental validations of the method's effectiveness.

### Weaknesses:

1. The comparison with existing methods is severely lacking. Although this work highlights the important issue of whether large models are safety-consistent across different persona settings, it is possible that existing alignment methods have already addressed this problem (perhaps unknowingly). Therefore, evaluating how well current alignment methods address this issue is crucial for demonstrating the value of the proposed approach.

2. The graph optimization and lineage credit propagation in PLE may increase computational burden, affecting efficiency in practical applications. If possible, I hope the authors can provide insights into time and token consumption.

3. I am curious whether placing this persona setting in the system prompt or in memory would yield different effects, as this relates to how the method would perform in real-world scenarios.

---

> ### Author Rebuttal · Authors · 2026-03-30
>
> Thank you for your insightful comments. We have carefully addressed your concerns below and would appreciate your reconsideration if you find them adequately addressed.
>
> ### **Weakness 1 & Question**
>
> This is a good question. We have demonstrated strong performance against standard alignment baselines, including SFT and DPO. To further address your concern, we extend the comparison to a broader set of representative defenses, including *inference-time methods* (e.g., SmoothLLM [1] and LLM self-eval [2]), *fine-tuning-based safety models* (e.g., Self-RedTeam [3] and STAIR [4]). For STAIR, we adopt the open-source STAIR-Llama-3.1-8B-DPO-3 model. We conduct a unified comparison of all these methods against PICL under OOD persona-based jailbreak attacks.
>
> Our results show that existing defenses fail to generalize effectively to OOD persona-based jailbreak attacks. On Qwen2.5-7B-Instruct, PICL outperforms all baselines by a substantial margin (e.g., 2.6% on StrongREJECT and 1.5% on XSTest-contrast); the same trend holds on Llama-3.1-8B-Instruct. Across both models, PICL consistently achieves the lowest or near-lowest ASR on key benchmarks, demonstrating superior robustness. The results are presented in **Table 1** below.
>
> **Table 1. Robustness against OOD persona-based jailbreak attacks (lower is better).**
> |Method|StrongREJECT|WildGuardTest|XSTest-contrast|Harmbench|OR-Bench-toxic|
> |-|-|-|-|-|-|
> |Qwen2.5-7B-Instruct|0.773|0.211|0.445|0.633|0.753|
> | + SmoothLLM|0.387|0.142|0.270|0.273|0.394|
> | + LLM self-eval|0.252|0.093|0.285|0.275|0.256|
> | + Self-RedTeam|0.393|0.123|0.230|0.343|0.481|
> | **+ PICL (ours)**|**0.026**|**0.049**|**0.015**|**0.053**|**0.052**|
> |Llama-3.1-8B-Instruct|0.195|0.129|0.265|0.255|0.412|
> | + SmoothLLM|0.211|0.125|0.205|0.213|0.345|
> | + LLM self-eval|0.130|0.080|0.019|0.165|0.094|
> | + Self-RedTeam|0.187|0.115|0.125|0.178|0.251|
> | + STAIR|0.117|0.061|0.016|0.095|0.103|
> | **+ PICL (ours)**|**0.026**|**0.038**|**0.010**|**0.065**|**0.085**|
>
> We will incorporate these additional baselines and results in the revised manuscript to provide a more comprehensive empirical evaluation.
>
> **Reference**
>
> [1] Smooth-llm: Defending large language models against jailbreaking attacks. arXiv:2310.03684.
>
> [2] Llm self defense: By self examination, llms know they are being tricked. arXiv:2308.07308.
>
> [3] Chasing moving targets with online self-play reinforcement learning for safer language models.
> arXiv:2506.07468.
>
> [4] Stair: Improving safety alignment with introspective reasoning. arXiv:2502.02384.
>
> ### **Weakness 2**
>
> We provide a concrete characterization of the practical cost in terms of both time and token consumption. A full PLE run consists of 40 rounds. In each round, we query the Qwen3-Max API to generate 10 candidate personas (20–100 words each). Under this setup, a complete run takes approximately **4–5 hours** on four NVIDIA RTX 3090 GPUs, with an average API usage of **~100K tokens**.
>
> This cost is modest given the improvement in attack quality. Importantly, PLE incurs no additional computational overhead compared to Persona-GA. We will include these efficiency statistics in the revised manuscript for clarity.
>
> ### **Weakness 3**
>
> We explicitly evaluate this factor by placing the persona in either the user prompt or the system prompt, and re-running PLE under both settings. We report the ASR for persona placement in both the system prompt and the user prompt in **Table 2**, and present the ASR differences between system-level and user-level placement in Table 3 (see supplementary link: https://anonymous.4open.science/r/anonymous-result).
>
> The results show that prompt placement primarily affects weak seed personas, while *elite personas* discovered by PLE remain consistently effective across both positions, indicating that PLE does not overfit to a specific prompt slot.
>
> This leads to a clear practical implication: persona-based vulnerabilities persist regardless of whether the persona is injected via user-level context or higher-priority system instructions.
>
> **Table 2. Attack transferability of personas in different positions on Qwen2.5-7B-Instruct.**
> |Persona|Position|StrongREJECT|WildGuardTest|XSTest-contrast|Harmbench|
> |-|-|-|-|-|-|
> |initial persona|system prompt|0.285|0.264|0.195|0.301|
> ||user prompt|0.114|0.124|0.082|0.164|
> |elite persona|system prompt|0.925|0.346|0.824|0.721|
> ||user prompt|0.936|0.316|0.845|0.748|
>
> In our experiments, we primarily adopt the user prompt, as it more closely reflects real-world jailbreak scenarios where adversaries manipulate user inputs to influence model behavior.

---

> > ### Author Rebuttal · Reviewer_6JXK · 2026-04-04
> >
> > Thanks for the authors' detailed rebuttal, my concern resolved, I will increase my score to accept:)

---

> > > ### Author Response · Authors · 2026-04-04
> > >
> > > We sincerely appreciate your encouraging follow-up and are delighted to hear that our rebuttal has helped clarify the paper. We are also deeply grateful for your positive assessment and for recognizing the value of our clarifications.

---

### Decision · Program_Chairs · 2026-04-30

**Decision:**

Accept (regular)

**Comment:**

The paper introduces Persona Invariant Alignment (PIA) to combat persona-based jailbreak attacks. PIA is an adversarial self-play framework that consists of Persona Lineage Evolution (PLE) as the attack mechanism to discover high-risk adversarial personas and Persona-Invariant Consistency Learning (PICL) as the defense mechanism that enforces safety under persona perturbations. Authors put their framework to test and show the effectiveness of their attack and defense.

Reviewers acknowledged the importance of the work and the clarity of the paper. The paper includes a coherent theoretical and empirical discussion. The empirical results are strong. One reviewer raised methodological novelty concerns along with their concern regarding shallow alignment. However, authors demonstrate strong generalization to OOD attacks. All other reviewers were supportive of the paper and thought that the paper addresses a timely and important issue. Authors provided a comprehensive rebuttal and included results for utility evals, including additional baselines, and models, efficiency results which resolved most reviewer comments. It is highly recommended for the authors to include these new results and discussions in the paper along with an ethics statements about limitations, ethical risks, and efforts to reduce these risks in an ethics statement section.